# Neuronal Menin Overexpression Rescues Learning and Memory Phenotype in CA1-Specific α7 nAChRs KD Mice

**DOI:** 10.3390/cells10123286

**Published:** 2021-11-24

**Authors:** Shadab Batool, Basma Akhter, Jawwad Zaidi, Frank Visser, Gavin Petrie, Matthew Hill, Naweed I. Syed

**Affiliations:** 1Hotchkiss Brain Institute, University of Calgary, Calgary, AB T2N 4N1, Canada; shadab.batool@ucalgary.ca (S.B.); basma.akhter@ucalgary.ca (B.A.); Fvisser@ucalgary.ca (F.V.); gavin.petrie@ucalgary.ca (G.P.); mnhill@ucalgary.ca (M.H.); 2Department of Neuroscience, University of Calgary, Calgary, AB T2N 4N1, Canada; 3Cumming School of Medicine, University of Calgary, Calgary, AB T2N 4N1, Canada; jawwad.zaidi@ucalgary.ca; 4Department of Cell Biology and Anatomy, University of Calgary, Calgary, AB T2N 4N1, Canada

**Keywords:** learning and memory, nicotinic cholinergic receptors, menin, synaptogenesis, extracellular activity, brain connectivity

## Abstract

The perturbation of nicotinic cholinergic receptors is thought to underlie many neurodegenerative and neuropsychiatric disorders, such as Alzheimer’s and schizophrenia. We previously identified that the tumor suppressor gene, *MEN1*, regulates both the expression and synaptic targeting of α7 nAChRs in the mouse hippocampal neurons in vitro. Here we sought to determine whether the α7 nAChRs gene expression reciprocally regulates the expression of menin, the protein encoded by the *MEN1* gene, and if this interplay impacts learning and memory. We demonstrate here that α7 nAChRs knockdown (KD) both in in vitro and in vivo, initially upregulated and then subsequently downregulated menin expression. Exogenous expression of menin using an AAV transduction approach rescued α7 nAChRs KD mediated functional and behavioral deficits specifically in hippocampal (CA1) neurons. These effects involved the modulation of the α7 nAChR subunit expression and functional clustering at the synaptic sites. Our data thus demonstrates a novel and important interplay between the *MEN1* gene and the α7 nAChRs in regulating hippocampal-dependent learning and memory.

## 1. Introduction

Cholinergic transmission in the central nervous system (CNS) serves multiple functions ranging from development [1,2], modulation of excitatory [3] and inhibitory [4] transmission [5,6], central processing of pain [7], food intake, anxiety [8] to learning and memory [9]. Activation of cholinergic synapses in the mammalian CNS mediates synaptic modulation and plasticity [3], underlying learning and memory, specifically in the hippocampus [10,11]. Amongst these nicotinic cholinergic receptors (nAChRs), the α7 subunit has attracted significant attention due to its unique properties, such as high Ca^2+^ permeability-induced membrane depolarization [12,13], and its role in cognition, memory, immunity, inflammation and neuroprotection [14,15]. Specifically, α7 nAChRs’ role in hippocampus-specific learning and memory and its underlying mechanisms have been the focus of ongoing research [16,17]. Moreover, the perturbation of α7 receptor is linked to several neurological disorders, such as schizophrenia (SZ)[18,19] and Alzheimer’s disease (AD) [20]. Despite the importance of various cholinergic functions, our understanding of the development, assembly and maintenance of neuronal nicotinic cholinergic receptors is, however, limited due to its widespread distribution and structural/functional diversity in the CNS [21]. Furthermore, although the molecular determinants of cholinergic synaptogenesis at the neuromuscular junction are well characterized [22,23,24], the mechanisms by which various scaffolding molecules regulate their synapse-specific targeting and function in the brain remain undefined. Whereas Resistant to inhibitors of cholinesterase (RIC-3) protein promotes the assembly and trafficking of α7 nAChRs [25,26], it is not present in all neurons expressing α7 subunit though [27], nor does it regulate receptor protein expression [25]. Another study has highlighted the role of Src family kinases (SFKs), notably Src and Fyn, in the cholinergic synapse formation and clustering in an invertebrate model [28]. NACHO, an auxiliary protein, on the other hand, acts as an essential chaperone for α7 nAChRs [29], but its function is confined primarily to receptor assembly and insertion, and not their targeting at specific synaptic sites involved in learning and memory. We had earlier identified the Multiple Endocrine Neoplasia type I (*MEN1*) gene and its encoded protein product, menin, as a regulator of nAChRs at specific synaptic sites in both invertebrate [30] and vertebrate [31] neurons.

*MEN1* is an extensively studied tumour suppressor gene [32], with well-defined functions in the endocrine tissue [33]. Menin is a multifunction scaffolding protein that regulates cell–cell interaction, extracellular and intracellular signalling cascades and interactions, nuclear transcription [34], etc. Whereas menin is ubiquitously expressed in CNS [35], its function is tissue-specific [36]. Recent studies from our laboratory [30,31] and others [37] have shed light on menin’s role in cholinergic synaptogenesis [38], synaptic plasticity [39], cognition [40] and depression [41] in the mammalian CNS. A recent study from our group characterized the spatiotemporal patterns of menin expression during development, concomitant with the nAChRs specific subunits and other elements of the synaptic machinery [42]. Taken together, these studies demonstrate that the nAChRs targeting specific synaptic sites is tightly coupled with menin function, though a reciprocal nature of this interaction is yet to be demonstrated. Although these studies highlight the role of menin in nAChRs synaptogenesis and subunit-specific clustering, the interdependence of α7 nAChRs with menin protein and its effect on learning and memory remains unknown. In this study, we posed several important questions to further decipher the role of menin in nAChRs regulation and vice versa: (i) Does there exist a feedback mechanism that interplays between menin and the α7 nAChRs? (ii) Does menin impact α7 nAChRs-mediated roles in learning and memory?

Here, we employed both in vitro and in vivo mouse models to decipher potential interdependence between menin and α7 nAChRs at the molecular, as well as the functional, level. We demonstrate for the first time that using AAVs containing an shRNA specifically targeting α7nAChRs for KD significantly alters menin protein expression. Furthermore, *MEN1* reintroduction, also using AAV for exogenous transgene expression, restored learning and memory deficits in the α7 nAChR KD mice. Taken together, our data highlight a reciprocal interdependence between α7 nAChRs and menin and underscore the importance of these two proteins in hippocampal-dependent learning and memory.

## 2. Methodology

### 2.1. Animals: Brain Slices and Neuronal Cell Culture

All guidelines, protocols and procedures regarding animal handling, care and use were followed as per the University of Calgary and Canadian Council on Animal Care animal care certification(Ethical code number: REB19-1175_REN1; approval date: 4 October 2020). Adult C57BL/6 (Charles River, Calgary, AB, Canada) mice ranging from 2 to 4 months of age were used for experiments involving immunohistochemistry (IHC) of brain slices. The (adult) mouse brains were fixed with 4% paraformaldehyde overnight at 4 °C. The fixed brains were then subjected to 2% sucrose solution (submerged), snap-frozen with Optimal cutting temperature (OCT) compound block and stored at −80 °C. The cryostat was used to prepare 16–18 µm brain slices, at −20 °C as previously mentioned [43].

Pregnant female mice with E18 aged embryos were anesthetized with isoflurane and sacrificed by decapitation, and their brains were harvested for neuronal culture experiments. Hippocampal tissue was isolated from E18 embryos in solution (1 × HBSS containing 10 mM HEPES; 310 mOsm, pH 7.2), and treated with enzyme (Papain (50 U/mL) in 150 mM CaCl_2_, 100 µM L-cysteine and 500 µM EDTA in neurobasal medium (NBM)) for 20 min at 37 °C with 5% CO_2_. NBM supplemented with 4% FBS, 2% B27, 1% penicillin-streptomycin and 1% L-glutamine (GIBCO) was given 3× to wash out the enzyme. Using a trituration technique with fire-polished glass Pasteur pipettes, neurons were then dissociated and plated onto glass coverslips at a density of around 900 cells/mm^2^ for achieving lower density cultures maintained in NBM supplemented, as aforementioned. The coverslips used were washed previously with nitric acid and coated with poly-D-lysine (30 µg/mL; Sigma Aldrich, Oakville, ON, Canada) and laminin (2 µg/mL; Sigma Aldrich, Oakville, ON, Canada) in costar 12-well plates (VWR). The neuronal culture media (about 50%, was replaced with NBM supplemented with 2% B27, 1% penicillin-streptomycin and 1% L-glutamine on day 2 of cultures and the neurons were maintained throughout at 37 °C with 5% CO_2_. The neuronal media was changed every consecutive day to maintain the neuronal growth. 

### 2.2. Immunocytochemistry and Immunohistochemistry

Immunocytochemistry (ICC) and IHC assays were employed to label proteins of interest as described previously [42]. Neuronal cultures were fixed on DIV 3, 7, 10, 14 and 20, respectively, with 4% paraformaldehyde and 0.2% picric acid (Sigma Aldrich) in 1 × PBS and permeabilized for 1 h with an incubation medium (IM) containing 0.5% Triton and 10% goat serum in 1 × PBS. Negative controls were performed to test the specificity of the antibodies, as described previously [31]. To label proteins of interest, primary antibodies (menin C-terminal epitope (Bethyl Laboratories, A300–105A, Montgomery, TX, USA); menin C-terminal epitope (Santa Cruz, SC-374371, Dallas, TX, USA); α-neurofilament (Novus Biologicals, NB300-222, Littleton, CO, USA); α-synaptotagmin (EMD Millipore, MAB5200); α-PSD-95 (Antibodies Incorporated, 75-028, Davis, CA, USA), were used at 1:500 in IM for 1 h. Secondary antibodies (Alexa Fluor 488, 568, or 680 conjugated goat α-rabbit, α-mouse or α-chicken (Invitrogen, Waltham, MA, USA)) were used at 1:100 in IM for 1 h. α7-nAChR were labelled with Alexa Fluor 555 conjugated α-Bungarotoxin (Invitrogen, B35451) and/or Alexa Fluor 488 and/or Alexa Fluor 647 at 2 µg/mL in IM for 1 h. Neuronal cultures were then subjected to three 15 m washes in 1x PBS after each incubation at room temperature. Cells were mounted using ProLong Gold antifade reagent with DAPI (Invitrogen).

For brain slices, sections were exposed to freshly made 0.3% hydrogen peroxide in 0.1% sodiumazide for 15 m to block any endogenous peroxidase activity to avoid background labelling of blood vessels. Slices were then subjected to heat mediated antigen retrieval step in Sodium citrate buffer for 10 min and washed with 1XPBS for 15 min. The rest of the protocol for labelling tissue was the same as mentioned above. The specificity of the antibodies used in this study has previously been confirmed using western blot analysis [31]. All antibodies were, however, further optimized on sagittal and coronal 16 μm thick adult mouse brain slices in the AP Research Lab of Alberta Precision Labs, using high precision, reliable and automatic methods, as described above.

### 2.3. Quantitative PCR (qPCR)

Total RNA was isolated from adult C57BL/6 mice (injected with α7 shRNA AAV and α7 scrambled AAV) hippocampus (CA1 region in some cases) using the RNeasy micro kit (QIAGEN, ON, Canada) according to the manufacturer’s instructions. Using the quantitative reverse transcription kit (Superscript Vilo, Thermofischer, Waltham, MA, USA), reverse transcription was performed. For the negative control groups, all components except the reverse transcriptase were included in the reaction mixtures. A LUNA kit was used for qPCR assay, and the primers used were directed to a region of 80–120 bases, and the Human glyceraldehyde 3-phosphate dehydrogenase (GAPDH) gene was used as the housekeeping (reference) gene. Control reactions and those containing cDNA from α7 scrambled cultures and/or hippocampi, α7 KD cultures and/or hippocampus were performed with 1 ng of template per reaction. The running protocol extended to 45 cycles consisting of 95 °C for 5 s, 55 °C for 10 s and 72 °C for 8 s using the QuantStudio setup. The specificity of all PCR reactions was checked by dissociation standard curve analysis graph plot, whereas the assay validation was confirmed by testing serial dilutions of pooled template cDNAs, suggesting a linear dynamic range of 2.8–0.0028 ng of the template. Efficiency values for qPCR primers ranged between 85% and 110% (R^2^ = 0.97–1.00). The expression of genes in different treatment groups relative to GAPDH was determined using the Thermofischer QuantStudio™ 3D AnalysisSuite™ software, version 3.1.6.

### 2.4. Confocal Microscopy

Images were obtained using confocal microscopy [44] at 60× magnification on neuronal cultures and brain slices (16 µm), as previously described [42], and fluorophores were excited with 402, 488, 568 and 680 laser wavelengths and emissions collected through 450/50, 525/50 and 700/75 filter cubes for different samples. All imaging parameters, including the field of view size, laser intensity and channel gains, were kept strictly constant amongst relevant samples. Images were collected from 9–12 samples prepared from independent culture sessions. All microscope settings are shown (see Appendix A). The fluorescence intensity of all antibodies was measured and quantified using IMAGE J. The fluorescence range was from 0 to 250, where AU < 20 was considered background signal, 20 < AU > 50 was moderate and values >50 were deemed strong throughout the quantification of relevant samples. 

### 2.5. AAV Production and Transduction of Neuronal Cultures

#### α7 shRNA AAV and α7 shRNA Scrambled *MEN1* Encoding AAV and GFP Only AAV

Small hairpin (sh)RNA-encoding constructs were designed targeting the mouse CHRNA7 gene, which encodes α7 subunits of the nicotinic acetylcholine receptors using the splashRNA design tool [45]. The sequences with the highest “splash” scores were selected for experimental use (see Table 1).

### 2.6. Plasmid Construction and AAV Packaging

The shRNA sequences were incorporated into DNA oligos and were used to PCR amplify the U6 promoter of pAAV-U6-shRNA-CAG-tdTomato using Kapa Hifi DNA polymerase (Roche) according to the manufacturer’s instructions. The resulting PCR product was subcloned into the vector that had been digested with MluI and HindIII (see Appendix A). The correct sequence and insertion of the shRNA sequence were verified by DNA sequencing.

The mouse *MEN1* cDNA was obtained by ordering mammalian gene collection (MGC) Clone ID 4189611 from Horizon Discovery (NCBI accession NM_001168490.1). The cDNA sequence was PCR amplified and inserted in place of cre recombinase at the BspEI and HindIII restriction sites in pENN-AAV-hSyn-HI-eGFP-cre (a gift from James M. Wilson (Addgene plasmid #105540; RRID: Addgene_105540)) using the NEBuilder HiFi DNA assembly kit (New England Biolabs, Ipswich, MA, USA) to generate pAAV-hSyn-HI-eGFP-*MEN1*. The GFP only AAV had the same backbone, but no gene sequences were inserted (see Appendix A).

AAV viral vectors containing the AAV9 capsid were generated using the methods described previously by Challis et al. Briefly, 293FT cells (Thermofisher) were grown to ~90% confluency in Corning hyperflasks (Corning, New York, NY, USA) and co-transfected with 129 µg pHELPER (Agilent), 238 µg pAAV 2/9n rep-cap plasmid (pAAV2/9n was a gift from James M. Wilson (Addgene plasmid #112865; RRID: Addgene_112865) and 64.6 µg equimolar mixtures of pAAV-U6-shRNA-CMV-tdTomato or pAAV.hSyn-HI-eGFP-*MEN1* using the PEIpro transfection reagent (Polyplus, New York, NY, USA). AAVs were precipitated from media harvested after 3 days and 5 days using 40%PEG/2.5 M NaCl and the pooled cells were harvested after 5 days in buffer containing 500 mM NaCl, 40 mM Tris Base and 10 mM MgCl_2_. The lysate was incubated with 100 U/mL salt-active nucleases (Arcticzymes, Tromso, Norway) at 37 °C for 1 h and then centrifuged at 2000× *g* for 15 min. AAV was purified from the resulting lysate using an iodixanol step gradient containing 15, 25, 40 and 60% iodixanol in optiseal tubes (Beckman, Brea, CA, USA) followed by centrifugation at 350,000× *g* using a Type 70 Ti ultracentrifuge rotor (Beckman). Following centrifugation, the AAVs were harvested from the 40% layer using a 10 cc syringe and 16-gauge needle, diluted in 1XPBS containing 0.001% pluronic F68 (Gibco, Grand Island, NE, USA) and filtered using a 0.2 um syringe filter. The AAVs were concentrated and buffer-exchanged by 5 rounds of centrifugation using Amicon Ultra-15 100 kDa molecular weight cut off centrifugal filter units (Millipore, Burlington, MA, USA). The titer was determined using the qPCR Adeno-Associated Virus Titration kit (Applied Biological Materials), and the purity was verified by SDS-PAGE and total protein staining using instant blue reagent (Expedeon, Cambridge, UK).

Mouse hippocampal cultures were transduced with scrambled shRNA or α7 shRNA-encoding AAV on DIV 1 by spinoculation (2 m at 2000 rpm) using a multiplicity of infection of ~0.2, and the media was changed after 24 h. tdTomato fluorescence was observed after ~48–72 h, and the transduction efficiency of mouse hippocampal neurons was estimated to be ~90–94%. α7 nAChR KD was confirmed by qPCR and ICC and IHC.

### 2.7. Stereotaxic Injections in C57BL/6 Mice

Adeno-associated virus (AAV) (200 ng/μL) was administered bilaterally in the CA1 region of C57BL/6 mice through stereotaxic injections, as previously described [46]. The coordinates used for injections were 2 mm behind bregma, 2 mm lateral and 1.6 mm below the dural surface for CA1 [47]. The injections were performed with glass micropipettes with a tip diameter ranging between 10 and 20 μm and injected slowly with a pressure-injection system (see Appendix A). The animals were given post-surgery care for 5 days, where they were monitored for any discomfort or changes in diet or movement (see Appendix A). The recovery time post-injection ranged between 3 and 5 weeks, after which the animals were either sacrificed for the validation of the model or were subjected to a contextual fear conditioning behavioural test.

### 2.8. Contextual Fear Conditioning Behaviour Test

Contextual fear conditioning (learning and memory test specific to the hippocampus) was conducted in a chamber with plastic walls and a metal rung floor. The internal dimensions of the chamber were approximately 17 × 17 × 25 cm for mice. The mice behaviour was recorded by a digital video camera directly mounted above the conditioning chamber. The setup was acquired from the ANY-maze Stoelting Co. (Wood Dale, IL, USA) Fear Conditioning System, which automatically detected and quantified the freezing behaviour. On Trial Day 1, the animals were placed into the conditioning chamber and habituated to their surroundings for 2 min. Following the habituation, the mice received 0.5 mA shocks of 1 s duration every 2 m. Once the trial was completed, the animals were returned to their home cage, and the chamber was cleaned with 70% ethanol after each trial. 

On trial day 2, the mice were placed in the identical environment (chamber) and were subjected to behavioural analysis as day 1, but the shock was not presented. Each minute, the software recorded the freezing score (freezing episodes and freezing time; freezing percentage) for each animal. After completion of the task, the mice were returned to their home cages. The results were exported in excel and analyzed using PRISM as aforementioned. 

### 2.9. Experimental Design and Statistical Analysis

To ensure that all results were reproduceable and replicable, data sets were derived from ≥8 independent experiments, using samples from ≥12 independent cell culture preparations or tissue collected from ≥8 animals. The resource equation method was used to determine the minimum sample sizes for quantitative data. IMAGE J (NIH) software was used for image processing and fluorescence intensity unbiased blinded by acquisition file number. For brain slices, an equal area for the region of interest (ROI) was selected randomly (independent of the size of the tissue), *n* ≥ 12 every slice per region to minimize biases. The quantification tools were kept constant for all tissues amongst relevant samples.

Statistical analyses were performed using Prism8 version 8.3.1 Graph pad software. The data distribution was analyzed with the D’Agostino and Pearson test of normality [48], Bartlett’s test for homoscedasticity and parametric (*p* > 0.05) or non-parametric (*p* < 0.05) statistical tests were used as appropriate. Differences in fluorescence intensity for ICC were assessed with the one-way ANOVA followed by a post-hoc Tukey test on IMAGE J, Java 1.8. Significant differences in relative gene expression from qPCR from relevant samples on different DIVs were determined using one-way ANOVA followed by a post-hoc Tukey test and Dunnett’s multiple comparison test amongst relevant samples. Significant differences in the degree of colocalization test were determined by a one-sample *t*-test.

## 3. Results

### 3.1. Selective Hippocampal, Neuron Specific KD of α7 nAChRs Differentially Regulates the Expression of the MEN1 Gene during Synaptogenesis and Synaptic Maturation

Previous studies from our lab [30,31,42], and others [39], have shown the *MEN1* gene and its encoded protein’s roles in synaptogenesis [38], synaptic plasticity [31,49] and learning and memory [40]. Moreover, neuronal *MEN1* knockout mice exhibit learning and memory deficits [40]. However, it remains to be determined whether there exists reciprocal feedback between menin and α7 nAChRs in cholinergic synaptogenesis and function. To test this possibility, we sought to determine whether selective KD of α7 nAChRs alters the expression patterns of the *MEN1* gene and its encoded protein menin. 

To this end, we created a plasmid construct with U6-promoter-driven, α7 nAChR-specific shRNA or its scrambled control together with CMV-tdTomato and packaged it into AAV serotype 9, which has been shown previously to exhibit efficient transduction in mouse brain neurons [50] with the aim of impairing α7 nAChRs expression in the hippocampal neurons. Hippocampal neurons were dissected from C57BL/6, E18 mouse pups, and neuronal cultures were prepared for virus transduction. Primary hippocampal neurons were transduced with α7 shRNA AAV and its relevant scrambled control on day in vitro 1 (DIV 1). The transduced neurons were then imaged every day to monitor the AAV transduction using tdTomato fluorescence as an indicator of transduction efficiency (see Figure 1A,B). On DIV 10 (which represents the period of active synapse formation in cultures) and DIV 20 (the period of synapse maturation in culture) [51], RNA samples were collected from untreated controls, α7 scrambled AAV and α7 shRNA AAV for qPCR analysis and assayed for nAChRs neuronal subunits α2-7 and β2-4, as well as the *MEN1* gene. Our results demonstrate that α7 nAChRs scrambled AAV had no significant effect on the expression levels of any of the genes that we examined, including nAChRs neuronal subunits α2-7 and β2-4, as well as the *MEN1* gene (see Figure 1C,D; *n* = 5, *3* independent experiments each, see Appendix A) relative to untreated control on DIV 10–DIV 20. However, in α7 nAChRs shRA AAV transduced cultures on DIV 10, we observed a 4-fold upregulation of the *MEN1* gene, as well as a 2.5-fold upregulation of α5 nAChRs, whereas a significant 12.7-fold downregulation of α7 nAChRs (see Figure 1C, *n* = 5, three independent experiments each, see Appendix A). Interestingly on DIV 20, α7 nAChRs KD neurons exhibited about a 13.9-fold downregulation of the *MEN1* gene (see Figure 1D, *n* = 5, three independent experiments each, see Appendix A). These results suggest that there might exist a feedback loop between α7 nAChRs and the *MEN1* gene, and as such, underscore the importance of the *MEN1* gene regulation via subunit-specific, nicotinic cholinergic receptor function. 

Next, we used an ICC fluorescence intensity analysis to verify α7 nAChRs KD in α7 nAChRs shRNA transduced hippocampal neurons and its relevant control (scrambled AAV). Images were acquired from tdTomato-positive neurons using confocal microscopy to compare the α7 nAChRs expression in both groups from DIV 3, 7, 10 and 14, where fluorophore-tagged α-BTX was used to label α7 nAChRs. Our data confirmed a 93% reduction of α7 nAChRs in α7 nAChRs shRNA AAV transduced cultures compared to scrambled controls, which had no significant differences in α7 nAChRs expression compared to the untreated controls (see Figure 2A,B, *n* ≥ 30 each; DIV 3–14, representative images on DIV 10, see Appendix A). These data validated a subunit-specific KD of nAChRs α7 protein and indicated an underlying feedback loop that may potentially mediate an initial upregulation followed by downregulation of the *MEN1* gene following the α7 nAChRs shRNA-induced KD in neurons.

### 3.2. α7 nAChRs KD in Hippocampal Neurons Perturbs Menin Expression

Having established that α7 nAChRs KD induced the downregulation of the *MEN1* gene, we next sought to determine whether the expression patterns of menin protein were altered in α7 nAChRs KD neurons in vitro. To address this question, we labelled α7 nAChR KD neuronal cultures with an antibody that recognized an antigen close to the C-terminus of menin (C-menin antibody) on different DIVs, 3, 7, 10, 14 and 20, and quantified the fluorescent intensity using IMAGE J as previously shown [42]. We observed significant changes in the intensity of puncta exhibiting C-menin antibody fluorescence in α7 nAChRs shRNA-transduced tdTomato-positive neurons compared to its scrambled controls during the synaptogenic period (see Figure 3A; *n* ≥ 30 each; DIV 3–14, representative images on DIV 10, see Appendix A). The C-menin fluorescence intensity in these puncta, however, gradually decreased compared with their scrambled controls in matured neurons over time (see Figure 3B; *n* ≥ 30 each; DIV 3–14, representative images on DIV 20, see Appendix A). This reduction in the numbers of menin-expressing puncta and the timelines were consistent with the *MEN1* gene expression data presented in Figure 1.

### 3.3. α7 nAChRs KD in Hippocampal Neuronal Cultures Altered Synaptic Proteins Assembly at the Pre and Postsynaptic Sites

Menin colocalizes both at the pre and postsynaptic sites with SYT and PSD-95 along with the cholinergic machinery in the mouse hippocampus [42]. Neuron-specific *MEN1* knockout exhibits an altered synaptic expression of Synaptotagmin, SYT (presynaptic vesicle protein) and Postsynaptic density protein, PSD-95 [40]. Having established that menin is altered in α7 nAChRs KD neurons, we next sought to determine what would happen to the expression of other synaptic proteins involved in cholinergic synaptogenesis [31]. To address this question, we used α7 nAChR shRNA-transduced neuronal cultures and performed ICC. Specifically, we used a PSD-95 antibody to label the postsynaptic PSD-95, SYT-1 antibody to label presynaptic SYT and fluorophore-tagged α-BTX to label α7 nAChRs on DIV3, 7, 10 and 14, respectively. Consistent with our previously published data [40], the ICC results demonstrated a significant reduction in the puncta exhibiting PSD-95 fluorescence in α7 nAChR KD neurons from DIV 7 onwards (see Figure 4A,B, *n* ≥ 30 each; DIV 3–14, representative images on DIV 10, see Appendix A). In contrast to postsynaptic, the puncta-expressing presynaptic protein, SYT significantly increased in tdTomato-positive α7 nAChR KD neurons compared to our scrambled and untreated controls (see Figure 5A,B, *n* ≥ 30 each; DIV 3–14, representative images on DIV 10, see Appendix A). Taken together, these results suggest a menin-dependent perturbation of both pre and postsynaptic proteins and underscores the importance of the menin-induced α7 nAChRs role in the assembly of synaptic machinery.

### 3.4. α7 nAChRs KD Altered MEN1 Gene Expression and Synaptic Assembly in α7 nAChRs shRNA AAV (CA1 Specific) KD Mouse 

To test whether menin and α7 nAChRs expression are interdependent in the intact mouse hippocampus as observed in the in vitro neuronal cultures, we used two-month-old adult C57BL/6 mice and stereotaxically injected α7 nAChRs or scrambled control shRNA-AAV9 in CA1 coordinates (bilaterally in both hippocampi) (see Figure 6A). The weight and the feeding habits of the animals were closely monitored post-surgery, and there were no significant reported differences between the brain weight between α7 nAChRs KD mice compared to its relevant controls (See Appendix A). Four weeks post-injection, the animals were sacrificed, and the CA1 hippocampal region was dissected for RNA analysis followed by qPCR, where we assayed for nAChRs neuronal subunits α2-7, β2-4, SYT and PSD-95, as well as the *MEN1* gene. Our qPCR data confirmed the downregulation of α7 nAChRs, which validated their KD, specifically in the CA1 region. Additionally, our data also showed the downregulation of the *MEN1* gene in α7 nAChRs KD neurons, whereas the *MEN1* gene expression in the scrambled controls remained unchanged relative to the untreated controls (see Figure 6B, *n* = 5, three independent experiments each with 3X replicates, see Appendix A). Interestingly, in the α7 nAChR KD tissues, we also observed a downregulation of PSD-95 and an upregulation of the SYT gene, which further confirmed that the α7 nAChR KD did indeed perturb menin-associated synaptic assembly.

Having confirmed that the α7 nAChR and *MEN1* genes were indeed downregulated at the gene level, we next prepared brain slices to look for the expression levels of α-Bungarotoxin labelled nAChRs-α7 in the CA1 region using IHC assay. Our IHC results were consistent with the in vitro findings, confirming that indeed α7 nAChR expression was significantly reduced in α7 nAChR KD mice in the CA1 region (see Figure 6C and Figure 7C, *n* ≥ 25 each; adult mouse brain hippocampi, representative images, see Appendix A). Taken together, these data propose that an analogous mechanism of a feedback loop might exist between α7 nAChRs and menin protein in the hippocampal neuronal networks.

To examine whether menin protein was altered in vivo, as well in α7 nAChRs KD in the CA1 neurons, we employed an IHC assay to analyze IHC fluorescence intensity (IMAGE J) in the synaptic puncta. We labelled hippocampal brain slices with the C-menin antibody, as described earlier. The IHC results from the fluorescent intensity measurement of C-menin were consistent with our in vitro cell culture data (DIV 20), which exhibited decreased C-menin protein expression in α7 nAChR KD mouse brain slices (see Figure 7A,B, *n* ≥ 25 each; adult mouse brain hippocampi, representative images, see Appendix A). Taken together, our in vivo results were consistent with that of the in vitro, and together they underscore the importance of the *MEN1* gene regulation via subunit-specific, α7 nAChR function in the hippocampus. 

As shown in our in vitro α7 nAChR KD model (see above), both pre and postsynaptic proteins were perturbed in α7 nAChR shRNA-induced neurons; we next sought to investigate whether the synaptic assembly in the intact hippocampus was also altered owing to menin downregulation in α7 nAChR KD neurons. To address this question, we labelled hippocampal (CA1) brain slices with PSD-95 and SYT-1 antibodies, respectively (as aforementioned), and quantified the fluorescence intensity using IMAGE J, as previously established [42]. Intriguingly, we observed a significant reduction in PSD-95 expression in the α7 nAChR KD CA1 region compared to our scrambled controls (see Figure 7C and Appendix A, *n* ≥ 25 each; adult mouse brain hippocampi, representative images, see Appendix A) and a significant increase in the SYT-1 fluorescence labelled puncta in α7 nAChR KD CA1 neurons (see Figure 7D and Appendix A, *n* ≥ 25 each; adult mouse brain hippocampi, representative images, see Appendix A) consistent with our in vitro findings. Taken together, these data suggest the importance of menin-induced α7 nAChRs clustering in the assembly of synaptic machinery in an intact mouse brain.

### 3.5. Restoring Menin in a7 nAChRs KD Hippocampal Neurons Rescued the Expression of a7 nAChRs and Clustering at Synaptic Sites

Previous studies have shown that the KD of the *MEN1* gene using an shRNA approach in the hippocampal neuronal cultures results in a loss of α7 nAChRs clusters at the synaptic sites [31]. Our data presented here demonstrate that α7 nAChRs KD in neurons both in vitro and in vivo downregulate *MEN1* and its protein product, menin (as shown above). We next asked whether the clustering of α7 nAChR at the synaptic sites was directly dependent on menin expression at the synapses. To address this question, we designed a construct with a neuron-specific synapsin promoter driven by recombinant *MEN1*, which was N-terminally tagged with eGFP and packaged into AAV9 to selectively target α7 nAChR KD cultures. We overexpressed exogenous recombinant, GFP-tagged *MEN1* encoding AAV in α7 KD cultures by transducing the hippocampal neuronal cultures with *MEN1* AAV on DIV1 (see Appendix A). We also employed CA1-specific α7 nAChR KD mice to stereotaxically inject *MEN1* encoding AAV with the same coordinates, as aforementioned for our in vivo experimental design (see Appendix A).

First, we assayed IHC to label GFP-positive neurons with neurofilament to show that the expression of exogenous *MEN1* through AAV was only specific to neurons (see Figure 8A, *n* ≥ 18 each; adult mouse brain hippocampi, representative images, see Appendix A). We next sought to determine menin expression in *MEN1* AAV-expressed neurons. Our ICC results for menin labelling also exhibited an increase in menin protein fluorescence expression compared to α7 nAChR KD neurons (see Figure 8B, *n* ≥ 18 each; DIV 3–14, representative images on DIV 20, see Appendix A). Our IHC findings were consistent with the neuronal culture data, which exhibited augmented fluorescence intensity labelled menin puncta in the CA1 region of α7 nAChR + *MEN1* mice (see Figure 8D and Appendix A, *n* ≥ 18 each; adult mouse brain hippocampi, representative images, see Appendix A).

We next isolated RNA samples from the five groups α7 nAChR scrambled AAV, α7 nAChR shRNA AAV, α7 nAChR scrambled AAV+*MEN1*, α7 nAChR AAV+*MEN1*AAV and α7 nAChR AAV+GFP AAV only, respectively, and used a qPCR assay to determine the expression of α7 nAChR and the *MEN1* gene in all five groups of neuronal cultures’ samples and CA1 brain tissues. Intriguingly, our qPCR data from neuronal cultures and in vivo hippocampal CA1 slices demonstrated that an overexpression of recombinant *MEN1* in α7 KD neurons restored the expression of the α7 nAChRs and *MEN1* gene in α7 nAChR AAV+*MEN1*AAV samples compared to our α7 nAChR AAV KD neuron samples. However, in our α7 nAChR scrambled AAV+*MEN1*AAV, there was a significant upregulation of both *MEN1* and α7 nAChRs (see Figure 9A and Appendix A, *n* = 5, three independent experiments each, see Appendix A). These findings strongly support the prediction that the *MEN1* gene plays an important role in the transcriptional regulation and modulation of α7 nAChRs.

We then sought to determine if an overexpression of the *MEN1* gene could restore the expression of α7 nAChRs clusters at the synaptic sites. To this end, we used an ICC assay to label the five following groups; (1) α7 nAChR scrambled AAV, (2) α7 nAChR shRNA AAV, (3) α7 nAChR AAV+*MEN1*AAV, (4) α7 nAChR scrambled AAV+*MEN1* and (5) α7 nAChR AAV+GFP AAV, respectively, with α-Bungarotoxin labelled and c-terminal menin antibody to measure the expression of these proteins using fluorescence intensity. Our ICC results demonstrated that an overexpression of *MEN1* rescued α-Bungarotoxin-labelled α7 nAChRs in α7 nAChR KD neuronal cultures (see Figure 9B,C, *n* ≥ 18 each; DIV 3–14, representative images on DIV 20, see Appendix A). Nicotinic cholinergic receptors’ specific subunits of α7 nAChRs clusters were also restored in the CA1 region of α7 nAChR+*MEN1* mice, as shown by our IHC data (see Figure 10A and Appendix A, *n* ≥ 18 each; adult mouse brain hippocampi, representative images, see Appendix A).

Taken together, our findings suggest menin-dependent clustering of α7 nAChRs and underscore its importance in regulating α7 nAChRs assembly at the synaptic sites.

### 3.6. Overexpression of Exogenous Menin in the α7 nAChRs KD Mice Rescues Hippocampus Dependent Learning and Memory

Next, we sought to determine whether the loss of CA1-specific α7 nAChRs knockdown had any significant impact on learning and memory in these mice. To this end, we first performed a contextual fear conditioning assay (which is specific to hippocampus-dependent learning and memory) in α7 nAChRs knockdown mice (see Figure 10B). Our data from α7 nAChRs knockdown mice indicated significant deficits in learning and memory (no difference in freezing episodes between two days) (see Figure 10C, *n* = 15, see Appendix A). These data specifically demonstrated that the selective knockdown of nAChRs α7 in hippocampal neurons significantly affected learning and memory in freely behaving mice. To test whether exogenous *MEN1* encoding AAV could restore this learning and memory deficit in α7 knockdown mice, we then stereotaxically injected recombinant *MEN1*-encoding AAV in the same coordinates (CA1) as mentioned earlier in the α7 knockdown mice, four weeks post-surgery, and performed the same behavioural assay (contextual fear conditioning) on α7 nAChR shRNA AAV+*MEN1*AAV mice (see Figure 10C). Remarkably, our data showed significant improvement in learning and memory of α7 nAChR shRNA AAV+*MEN1*AAV mice compared to its relevant controls (significant increase in freezing episodes on day 2 than day 1) (see Figure 10C, *n* = 15, see Appendix A). The freezing percentage in all the four groups exhibited the same trend as observed in the freezing episodes (See Appendix A and Appendix A).

Taken together, the functional and behavioural data highlight the importance of *MEN1*’s role in α7 nAChR-dependent connectivity and learning and memory.

**Figure 10 cells-10-03286-f010:**
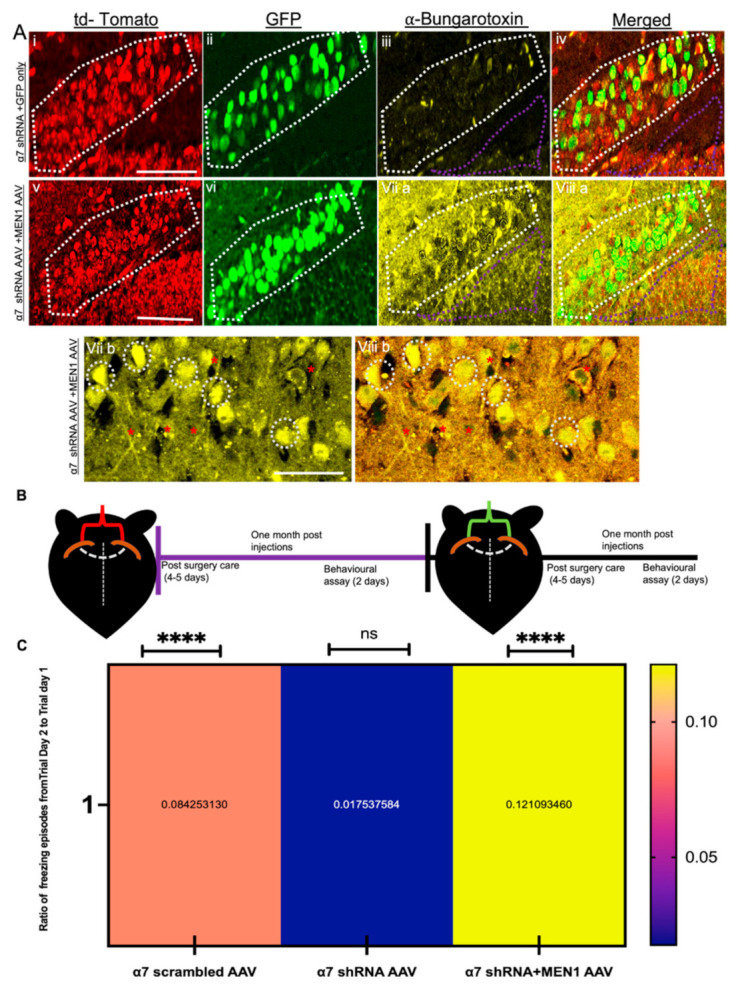
Overexpression of exogenous menin in the a7 nAChRs KD mice rescues hippocampus-dependent learning and memory**.** (**A**) IHC characterization of α7 KD+*MEN1* AAV hippocampal brain slices (**Bv**–**viii**), compared to scrambled controls (**Bi**–**iv**) co-transduced with GFP only AAV (*n* = 25 images, six independent samples, representative image). tdTomato-positive neurons (**Bi**,**v**) and GFP positive neurons (**Bii**,**vi**) labelled with α-bungarotoxin (**Biii**,**viia**,**b**) and merged (**Biv**,**viiia**,**b**), **Bviia,b** exhibits significantly increased expression of α7 nAChRs in tdTomato+GFP-positive hippocampal pyramidal neurons compared to the α7 KD+GFP only control (**Biii**).Panels viib and viiib display magnified images to display increased α7 nAChRs in hippocampal neurons (α7 KD+*MEN1* AAV). (**B**) Illustrative representation of experimental protocol timeline for bilateral stereotaxic injections in CA1 coordinates; red indicates α7 nAChR scrambled AAV, α7 nAChR shRNA AAV injections, green indicates α7 nAChR scrambled AAV+*MEN1*, α7 nAChR AAV+M*EN1*AAV and α7 nAChR AAV+GFP AAV viral administration. (**C**) Heat map for contextual fear conditioning test in three groups, representing the ratio of freezing episodes on day 2 compared to trial day 1. White dotted region indicates GFP-positive+tdTomato-positive pyramidal neurons. White dotted circles show increased expression of α7 nAChRs in α7 KD+*MEN1* AAV neurons. Purple dotted shape indicate synaptic puncta region. Red asterisks indicate α7 nAChRs puncta expression in tdTomato+GFP-positive neurites. Statistical significance (one-way ANOVA followed by Tukey’s multiple comparison test) **** *p* < 0.0001, ns *p* > 0.05. See Appendix A.

## 4. Discussion

Menin, the protein product encoded by the *MEN1* gene, has been extensively studied for its role as a tumour suppressor. However, in the last two decades, menin’s role in CNS specific to synaptogenesis [30], the regulation of synaptic plasticity [39], cognition [40] and depression [41] has come to light. Evidence from our lab demonstrates that menin’s role is specific to nicotinic cholinergic transcription, regulation and clustering at the synaptic sites [31,38,42]. These studies provided insights into menin’s role specific to the nAChR α7 subunit; however, it has only been shown in vitro, thus limiting their scope. The data presented in this study provides strong evidence that α7 nAChRs KD in hippocampal neurons invokes differential regulation of the *MEN1* gene and its encoded protein, menin. Additionally, for the first time, we demonstrated here that restoring menin expression in the α7 nAChRs KD neurons rescues α7 nAChRs clustering and improved hippocampus specific learning and memory [52].

Several approaches, including knockout mouse models, have been used to study α7 nAChRs’ function in the CNS [53,54,55]. α7 nAChRs are widely distributed in the whole brain where their expression is found both in neuronal and non-neuronal cells [9,56]; therefore, a global knockout model for α7 nAChRs limits the study for exploring its region-specific function. We sought to KD the α7 nAChRs (as previously shown [8]) specifically in the CA1 region to better understand the CA1-specific role of α7 nAChRs in learning and memory and its underlying molecular mechanisms. For this, we used the shRNA AAV approach [57], which has been shown to effectively KD the expression of the desired gene specific to the targeted region of interest. In our current study, shRNA against the CHRNA7 gene knocked it out in approximately 93% of hippocampal neurons, consistent with the successful AAV approaches used in other studies [57].

The hippocampus is a complex brain structure, which is the fundamental foci for learning and memory in the CNS [58,59], and its atrophy has been linked to memory impairment in AD patients [52].

Contextual fear conditioning, not cued fear conditioning (which is amygdala dependent) [60], is a hippocampus-dependent Pavlovian conditioning test [61], which has been used in past studies [62,63] for hippocampus-specific learning and memory [64]; therefore, we specifically tested for contextual fear conditioning test in our α7 KD model. Even though there are other behavioural tests to explore the role of the hippocampus in learning and memory [65,66], memories tested through these behavioural assays have been shown to have influences from other brain regions [67] as well. Recent studies have demonstrated differential roles of the hippocampal regions, *Cornu ammonus*, CA1 and CA3 in contextual learning and memory [68], whereas CA1 has been found to be necessary for encoding contextual learning and memory and its retrieval in mammals [69,70,71]. Cholinergic transmission modulates synaptic plasticity, which is required for learning and memory [9,10] specific to the hippocampus. Specifically, α7 nAChRs have been shown to play a role in cognition and learning and memory specific to the hippocampus [3,72]. In the present study, our data from the CA1-specific α7 nAChR KD mice demonstrate deficits in learning and memory using contextual fear conditioning test consistent with past findings, which emphasizes α7 nAChRs role in learning and memory [3,10]. Past studies have shown that menin deletion in neurons emphasizes its role in contextual learning and memory but not in cued learning and memory [40]. Intriguingly, our data showed that overexpressing *MEN1* in CA1-specific α7 nAChR KD neurons improves the contextual learning and memory phenotype compared to its relevant control. These findings suggest menin-dependent α7 nAChRs’ role in learning and memory, although an interesting find, the mechanisms underlying their association requires further detailed exploration. One limitation of our results could be that we only overexpressed exogenous menin specifically in neurons, whereas α7 nAChR-stimulated hippocampal-specific learning and memory are also affected by inputs from non-neuronal cells [61,73]. Our findings thus specifically accentuate the role of menin-dependent neuronal α7 nAChRs in learning and memory. Both menin and α7 nAChRs have been shown to be localized in glial cells [42,73]; therefore, the mechanisms underlying their roles in learning and memory through glial interaction need to be further explored. 

In the CNS, PSD-95 was previously thought to play a role in the assembly of glutamatergic receptors [74] and maturation of excitatory synapses specifically [75]; however, recent studies have highlighted its association with α7 nAChRs [76,77]. Intriguingly, our results from α7 nAChRs KD in primary hippocampal cultures, as well as the CA1 hippocampal brain slices, showed a reduction in the expression of PSD-95 puncta at the synapses, suggesting a bidirectional signalling that might be in play between these two proteins. Whereas PSD-95 at the postsynaptic sites was significantly reduced, we observed an upregulation of SYT (a presynaptic protein) in the α7 nAChRs KD cultures and hippocampal brain slice. Past studies on NDD pathophysiology have shown that PSD-95 is downregulated [78,79], whereas SYT is upregulated [80,81] in the hippocampus of dementia and AD models, consistent with our findings from α7 nAChRs KD. The *MEN1* knockout study by another group has shown that *MEN1* deletion in neurons upregulated SYT, whereas downregulated PSD-95 expression [40]. One possible reason for this perturbation of synaptic proteins in α7 nAChRs KD could be as a result of the ablation of menin expression in α7 nAChRs KD neurons, which could be a potential mechanism for AD pathophysiology; although this possibility is of significant interest, mechanisms underlying this phenomenon warrant further investigation.

Previous studies from our lab have shown that menin regulates α7 nAChRs transcription and clustering in invertebrate [30] and vertebrate neurons [31]. Knocking down the *MEN1* gene in hippocampal neurons perturbs the clustering and expression of nAChRs subunits, specifically α7 nAChRs [31], in vitro. In the present study, we have demonstrated, for the first time, that in the α7 nAChRs KD primary hippocampal neurons, *MEN1* is initially upregulated on DIV 10, whereas on DIV 20, it is significantly downregulated. These observations suggest that an initial upregulation of *MEN1* may be due to a compensatory feedback mechanism of the cell as menin protein is a transcriptional regulator and is involved in the transcription and clustering of α7 nAChRs. In in vitro studies, DIV 7–10 epitomizes the stage of active synaptogenesis and neuronal growth [51]. The ablation of α7 nAChRs provokes intensified menin expression, which suggests menin’s role in cholinergic synaptogenesis and compensatory action to promote neuronal growth [40]. Subsequently, on DIV 20, when the synapses are matured, and neuronal density is constant [51], our data shows that menin expression was significantly attenuated in α7 nAChRs knockdown hippocampal neurons. These findings could be indicative of the exhaustion of the intracellular machinery of the cell due to the overwhelming escalation in menin expression, which in turn might have led to negative feedback and downregulation of the menin protein [82]. As menin is involved in many intracellular signalling cascades and cell–cell interactions, its constant upregulation would disturb the cell’s homeostasis and compromise other signalling pathways [36]. Nicotinic cholinergic receptor-specific subunit α7 is highly permeable to Ca^2+^ ions [1,83], and their mutation has been shown to impair Ca^2+^ permeability of the neuron [84]. As shown by previous studies [30,85], *MEN1*′s regulation is through neurotrophic factor-mediated activity-dependent mechanisms. Taken together, we can speculate that long-term reduction of *MEN1* in α7 nAChRs KD neurons could be through Ca^2+^-dependent activity regulation. Although these speculations are of great interest, the mechanisms underlying these possibilities require further investigation.

Our data demonstrated that the restoration of α7 nAChRs by menin overexpression in the α7 nAChRs KD neurons, which suggests the possibility of menin’s involvement in the intracellular signalling of α7 nAChRs subunit-specific transcription, as menin has been reported to be a transcriptional regulator of α5 nAChRs [31], as well as other proteins in the cell [36]. Overall, these outcomes emphasize the multifaceted transcriptional networks underlying *MEN1* induction and gene targets that are transcriptionally activated or repressed by menin [86]. Our data also suggests menin’s role as a potential candidate as a scaffolding protein for α7 nAChRs amongst the other prospective chaperones [29,34,87] reported for α7 nAChRs, as the synaptic puncta clustering of α7 nAChRs augmented significantly in exogenous menin overexpressed α7 nAChR knockdown neurons. One of the explanations for *MEN1*-induced restoration of α7 nAChRs could be through the phenomenon of homeostatic synaptic scaling [88], which is a major feedback mechanism seen in other synaptic receptors as well, such as AMPARs [89,90] and mGLuR1 [91] for glutamatergic synapses. Cholinergic homeostatic synaptic plasticity [92] has been shown to induce regulatory responses via transcriptional activation of the *K**_v_**4/Shal* gene, indicating a receptor–ion channel system coupled for homeostatic modulations in neurons [93]. Taken together, these studies suggest that a similar mechanism of action could underlie *MEN1* induced restoration of loss of α7 nAChRs, hence re-establishing the cholinergic homeostasis. One important thing to consider is that adult neurogenesis [94,95] occurs in hippocampal neurons, and the increase in α7 nAChRs receptors and clustering could be due to the formation of new neurons; however, all these results were measured specifically in tdTomato-positive neurons (α7 nAChR knockdown), which filters any increase in nascent neurons in the hippocampus.

Overall, we have shown that α7 nAChR knockdown, both in vitro and in vivo, downregulates menin expression in neurons. Rescue by exogenous menin expression not only restores the α7 nAChRs transcription but also improves the α7 nAChR receptor clustering at the synaptic sites in the α7 nAChR knockdown neurons. The learning and memory impaired in α7 nAChR knockdown mice are rescued by menin overexpression specific to the CA1 region of the mouse hippocampus. Nicotinic cholinergic specific subunit α7 nAChRs are known to be perturbed in AD hippocampal pathology [20,96], schizophrenia [97] along with other synaptic proteins perturbations [96]. Ultimately, our results indicate a menin-dependent regulation of α7 nAChR expression and clustering, which might play a role in α7 nAChRs-related NDD pathophysiology.

## Figures and Tables

**Figure 1 cells-10-03286-f001:**
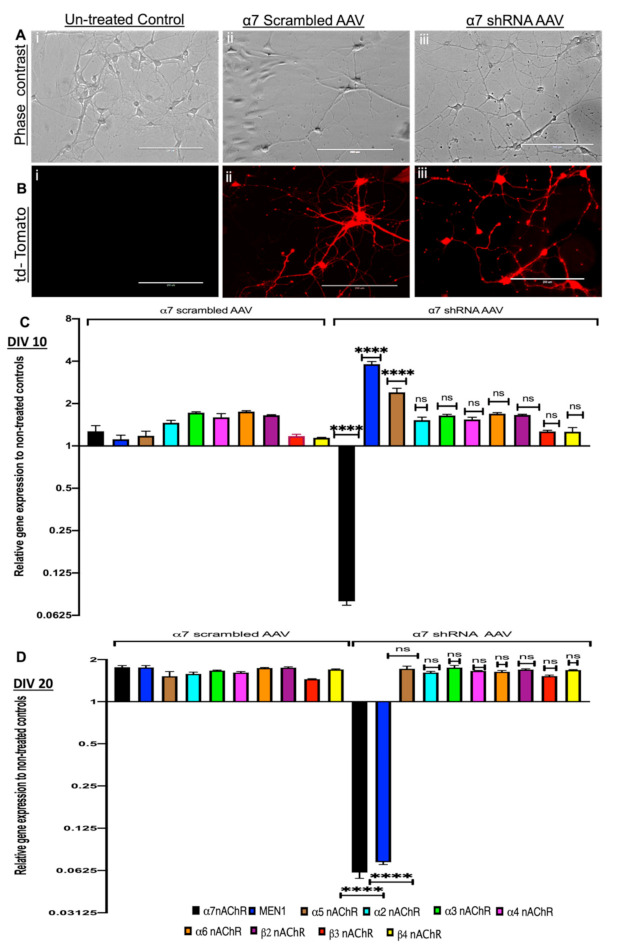
Selective KD of α7 nAChRs in the hippocampus differentially regulates the expression of Table 1. gene during synaptogenesis and synaptic maturation stage (in vitro). (**A**) live cell phase contrast (**i**–**iii**), (**B**) tdTomato fluorescence (**i**–**iii**); images of untreated control (**Ai**,**Bi**), scrambled control (**Aii**,**Bii**) and α7 shRNA AAV (**Aiii**,**Biii**) AAV transduced hippocampal cultures on DIV 7 (*n* = 21 images, 7 independent samples each, representative images). Scale bar 20 μm. (**C**,**D**) Summary data, fold change gene expression in hippocampal cultures on DIV 10 (**C**) and DIV 20 (**D**), relative to untreated control, determined by qPCR (*n* = 6, three independent experiments each, triplicate replicates). α7 KD upregulated *MEN1* expression initially and downregulated later in the synaptic maturation stage. Statistical significance (one-way ANOVA followed by Tukey’s multiple comparison test) **** *p* < 0.0001, ns *p* > 0.9999. See Appendix A.

**Figure 2 cells-10-03286-f002:**
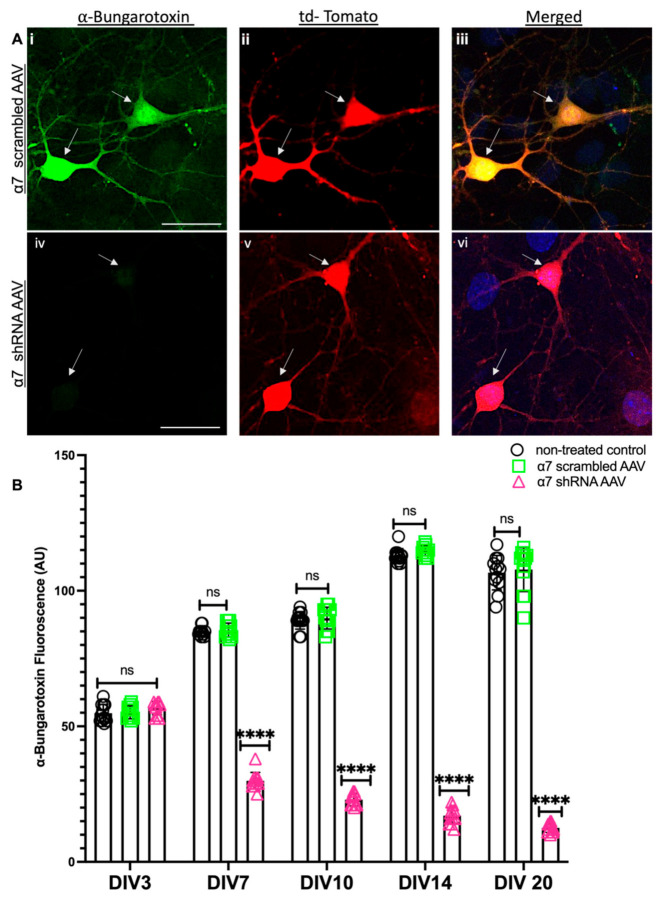
Reduced expression of Bungarotoxin labelled α7 nAChRs in α7 KD neurons (in vitro). (**A**) High-magnification (60×) confocal image of α7-nAChR shRNA AAV-transduced KD hippocampal neurons (**Aiv**–**vi**) compared to scrambled controls (**Ai**–**iii**) on DIV 10 (*n* = 30 images, six independent samples, representative image). tdTomato-positive cells (**Aii**,**v**) labelled with α-bungarotoxin (α7-nAChR) (**Ai**,**iv**) and merged (**Aiii**,**vi**), **Aiv** exhibits reduced expression of α7-nAChR in tdTomato-positive cells. Scale bars, (**Ai**–**iii**) 15 and (**Aiv**–**vi**) 15 μm. (**B**) Summary data, ICC characterization of α7-nAChR protein expression (normalized) in neuronal soma and neurites from scrambled shRNA and α7-KD hippocampal cultures compared to untreated controls (*n* ≥ 0 images, ≥6 independent samples, DIV 3, 7, 10, 14 and 20). Arrowheads indicate tdTomato-positive neurons. Statistical significance (one-way ANOVA followed by Tukey’s multiple comparison test) **** *p* < 0.0001, ns *p* > 0.9999. See Appendix A.

**Figure 3 cells-10-03286-f003:**
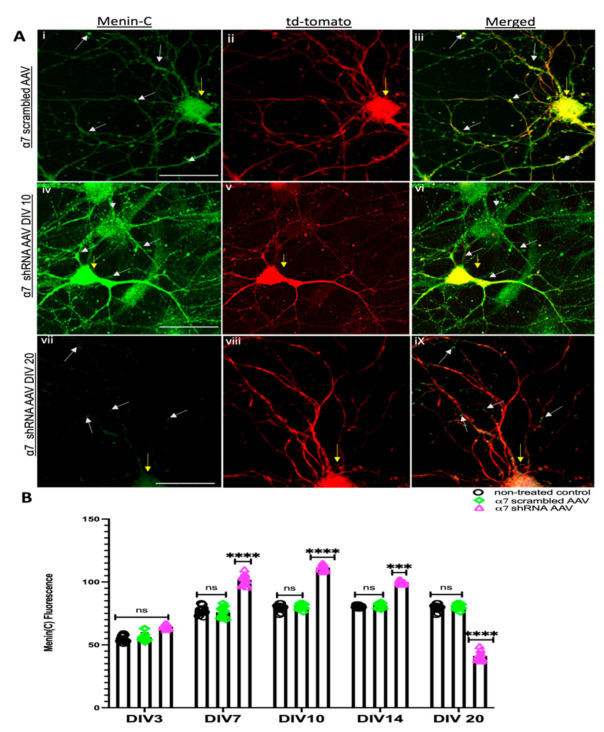
Menin protein expression increases initially and decreases later in α7 KD neurons (in vitro). (**A**) High-magnification (60×) confocal image of a α7 KD hippocampal neurons (**Aiv**–**ix**) compared to scrambled controls (**Ai**–**iii**) on DIV 10 and DIV 20 (**Avii**–**ix**), respectively (*n* = 30 images, six independent samples, representative image). tdTomato-positive cells (**Aii**,**v**,**viii**) labelled with α-C-terminal menin (**Ai**,**iv**,**vii**) and merged (**Aiii**,**vi**,**ix**). (**Aiv**) exhibits the augmented expression of α-C-terminal menin in tdTomato-positive cells on DIV 10, whereas (**Aviii**) displays a reduced expression of α-C-terminal menin expression and puncta in tdTomato-positive cells on DIV 20. Scale bars, (**Ai**–**iii**) 12, (**Aiv**–**vi**) 15 and (**Avii**–**ix**) 10 μm. (**B**) Summary data, normalized C-menin protein expression in neuronal soma and neurites from scrambled shRNA and α7-KD hippocampal cultures compared to untreated controls (*n* ≥ 15 images, ≥6 independent samples, DIV 3, 7, 10, 14, 20). Yellow arrowheads indicate tdTomato-positive neurons. White arrowheads indicate C-menin-positive puncta. Statistical significance (one-way ANOVA followed by Tukey’s multiple comparison test) **** *p* < 0.0001, *** *p* < 0.001, ns *p* > 0.9999. See Appendix A.

**Figure 4 cells-10-03286-f004:**
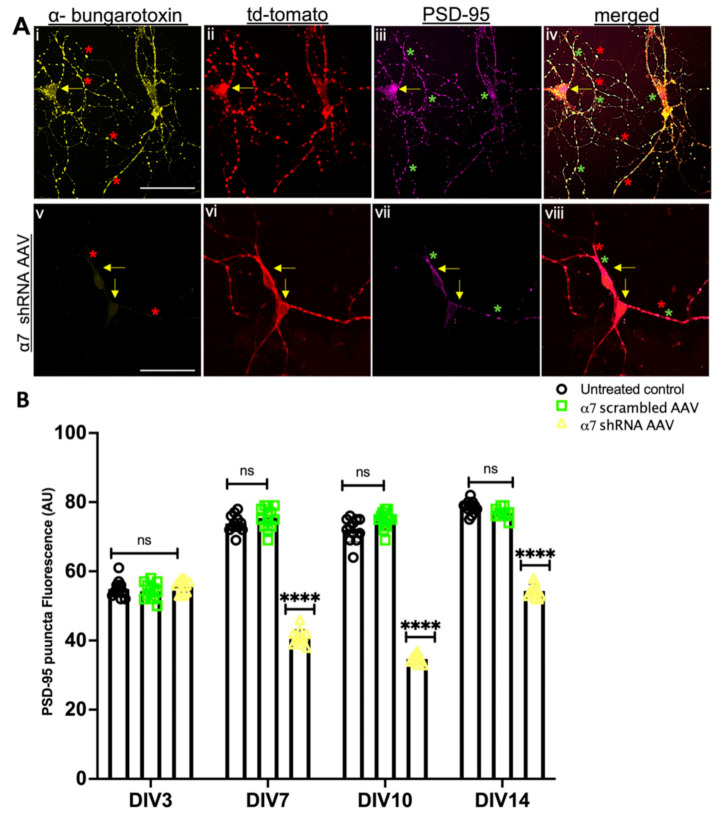
PSD-95 puncta expression reduces in α7 KD neurons (in vitro). (**A**) High-magnification (60×) confocal image of a α7 KD hippocampal neuron (**Av**–**viii**) compared to scrambled controls (**Ai**,**iv**) on DIV 10 (*n* = 30 images, six independent samples, representative image). tdTomato-positive cells (**Aii**,**vi**) co-labelled with α-bungarotoxin (α7-nAChR) (**Ai**,**v**) and PSD-95 (postsynaptic protein) (**Aiii**,**vii**) and merged (**Aiv**,**viii**). (**Avii**) exhibits the attenuated expression of PSD-95 in α7 KD hippocampal neurons compared to the scrambled control (**Aiii**). Scale bars, (**Ai**–**iv**) 25 and (**Av**–**viii**) 20 μm. (**B**) Summary data, normalized PSD-95 expression in neuronal soma and neurites from scrambled shRNA and α7-KD hippocampal cultures compared to untreated controls (*n* ≥ 25 images, ≥6 independent samples, DIV 3, 7, 10, 14 and 20). Yellow arrowheads indicate tdTomato-positive neurons. Red asterisks indicate bungarotoxin-positive puncta. Green asterisks indicate PSD-95 puncta. Statistical significance (one-way ANOVA followed by Tukey’s multiple comparison test) **** *p* < 0.0001, ns *p* > 0.9999. See Appendix A.

**Figure 5 cells-10-03286-f005:**
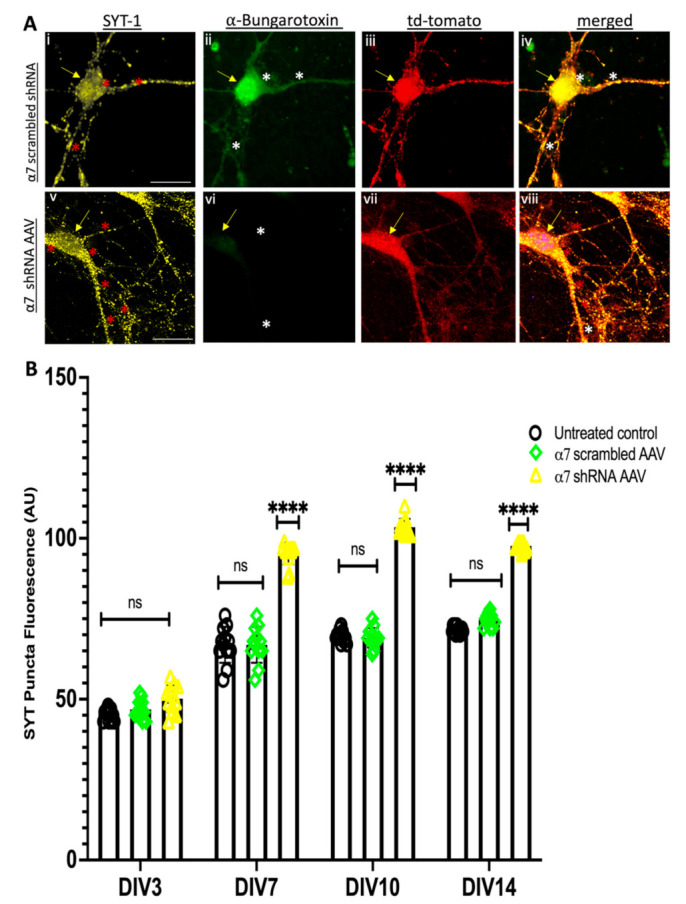
SYT-1 puncta expression increases in α7 KD neurons (in vitro). (**A**) High-magnification (60×) confocal image of a α7 KD hippocampal neurons (**Av**–**viii**) compared to scrambled controls (**Ai**,**iv**) on DIV 10 (*n* = 30 images, six independent samples, representative image). tdTomato-positive cells (**Aiii**,**vii**) co-labelled with α-bungarotoxin (α7-nAChR) (**Aii**,**vi**) and SYT-1 (presynaptic protein) (**Ai**,**v**) and merged (**Aiv**,**viii**). (**Av**) exhibits an augmented expression of SYT-1 in α7 KD hippocampal neurons compared to scrambled control (**Ai**). Scale bars, (**Ai**–**iv**) 15 and (**Av**–**viii**) 18 μm. (**B**) Summary data, normalized SYT-1 expression in neuronal soma and neurites from scrambled shRNA and α7-KD hippocampal cultures compared to untreated controls (*n* ≥ 25 images, ≥6 independent samples, DIV 3, 7, 10, 14 and 20). Yellow arrowheads indicate tdTomato-positive neurons. White asterisks indicate bungarotoxin-positive puncta. Red asterisks indicate SYT-1 puncta. Statistical significance (one-way ANOVA followed by Tukey’s multiple comparison test) **** *p* < 0.0001, ns *p* > 0.05. See Appendix A.

**Figure 6 cells-10-03286-f006:**
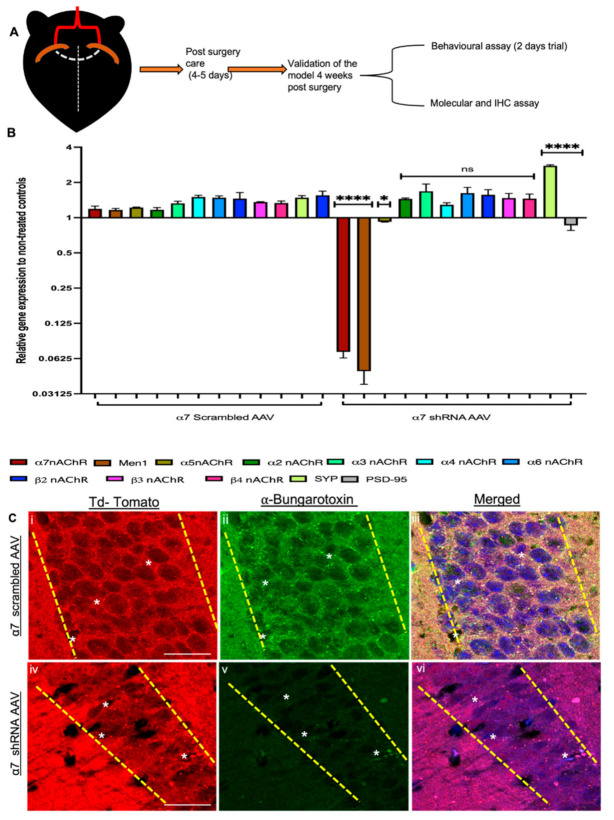
Downregulation of the *MEN1* gene in α7 KD mouse hippocampus (in vivo). (**A**) Illustrative representation of experimental protocol for bilateral stereotaxic injections in CA1 coordinates. (**B**) Summary data, fold change gene expression in α7-KD hippocampal tissue relative to untreated control, determined by qPCR (*n* = 6, four independent experiments each, triplicate replicates). Downregulation of the *MEN1* gene in α7-KD tissue. (**C**) IHC characterization of α7 KD hippocampal slices (**Civ**–**vi**) compared to scrambled controls (**Ci**–**iii**) 8 weeks post-surgery (*n* = 25 images, six independent samples, representative image). tdTomato-positive neurons (**Ci**,**iv**) labelled with α-bungarotoxin (α7-nAChR) (**Cii**,**v**) and merged (**Ciii**,**vi**), Av exhibits significantly reduced expression of α7-nAChR in tdTomato-positive hippocampal pyramidal neurons. Scale bars, (**Ci**–**iii**) 25 and (**Civ**–**vi**) 40 μm. Yellow dotted lines indicate tdTomato-positive pyramidal neuronal layers. White asterisks indicate bungarotoxin-positive puncta. Statistical significance (one-way ANOVA followed by Tukey’s multiple comparison test) **** *p* < 0.0001, * *p* < 0. 01, ns *p* > 0.9999. See Appendix A.

**Figure 7 cells-10-03286-f007:**
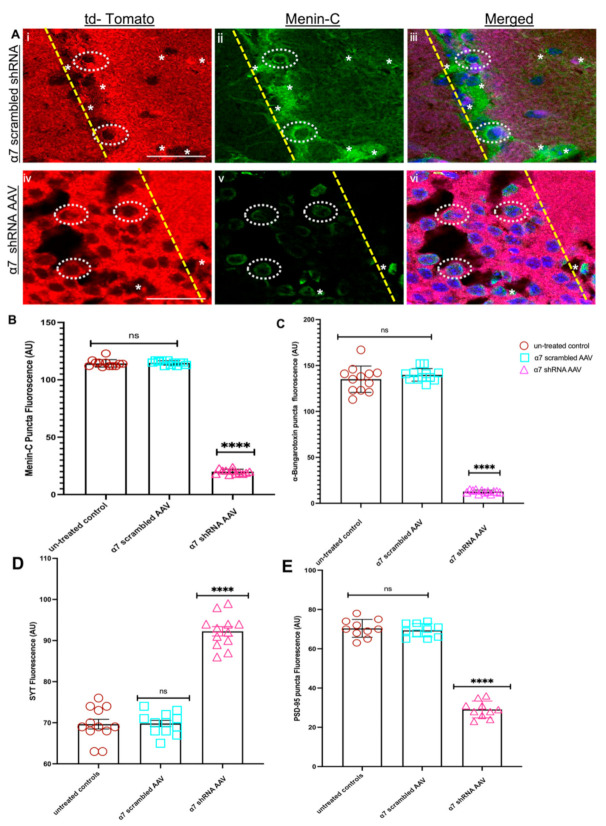
Menin protein is reduced in α7 KD mouse hippocampus (in vivo). (**A**) IHC characterization of α7 KD hippocampal slices (**Aiv**–**vi**) compared to scrambled controls (**Ai**–**iii**) 4 weeks post-surgery (*n* = 35 images, six independent samples, representative image). tdTomato-positive neurons (**Ai**,**iv**) labelled with α-C-terminal menin (**Aii**,**v**) and merged (**Aiii**,**vi**), Av exhibits significantly reduced expression of C-menin in tdTomato-positive hippocampal pyramidal neurons compared to scrambled control (**Aii**). (**B**) Summary data, normalized menin expression in neurons from scrambled shRNA and α7-KD hippocampal brain slices compared to untreated controls (*n* ≥ 25 images, ≥6 independent samples, 8 weeks old). (**C**) Summary data, normalized α7 nAChRs expression in neuronal soma and neurites from scrambled shRNA and α7-KD hippocampal brain slices compared to untreated controls (*n* ≥ 20 images, ≥6 independent samples, 8 weeks old). (**D**) Summary data, normalized SYT-1 expression in neurons from scrambled shRNA and α7-KD hippocampal brain slices compared to untreated controls (*n* ≥ 20 images, ≥6 independent samples, 8 weeks old). (**E**) Summary data, normalized PSD-95 expression in neurons from scrambled shRNA and α7-KD hippocampal brain slices compared to untreated controls (*n* ≥ 20 images, ≥6 independent samples, 8 weeks old). Scale bars, (**Ai**–**iii**) 12 and (**Aiv**–**vi**) 15 μm. Yellow dotted lines indicate tdTomato-positive pyramidal neuronal layers. White dotted circles menin expression in tdTomato-positive soma. White asterisks indicate menin expression in neurites. Statistical significance (one-way ANOVA followed by Tukey’s multiple comparison test) (Dunnett’s multiple comparison test) **** *p* < 0.0001, ns *p* > 0.9999. See Appendix A.

**Figure 8 cells-10-03286-f008:**
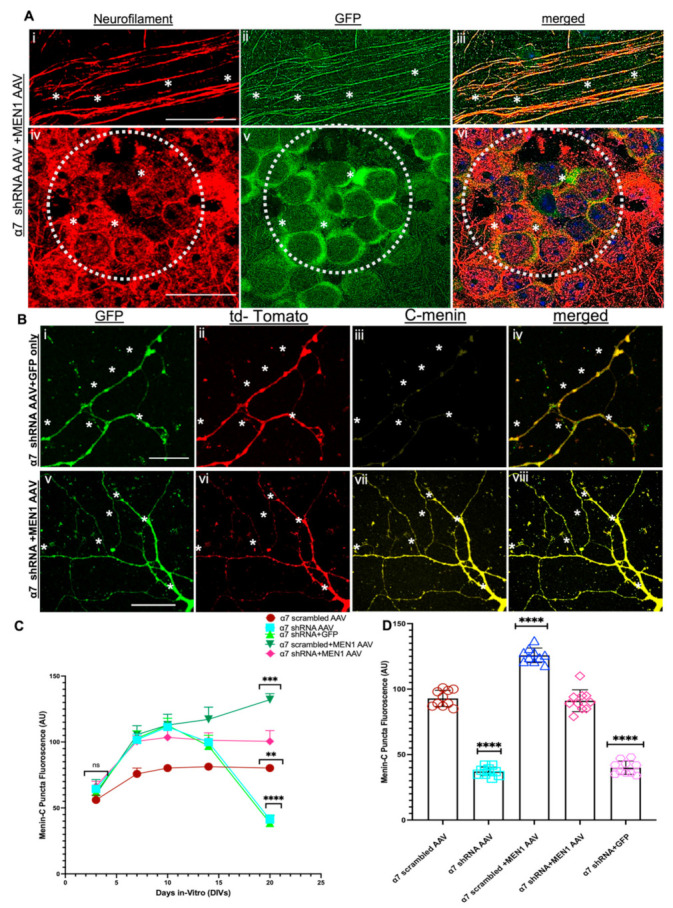
Exogenous expression of neuron-specific menin restores menin protein in α7 KD hippocampal neurons (in vitro). (**A**) IHC characterization of α7 KD hippocampal slices (**Ai**–**vi**) 12-week-old mice (*n* = 20 images, 5 independent samples, representative image). *MEN1* encoding AAV labelled GFP-positive neurons (**Aii**,**v**) labelled with neurofilament NFL (neuronal marker) (**Ai**,**iv**) and merged (**Aiii**,**vi**) exhibiting a strong degree of colocalization between GFP and NFL. (**B**) ICC characterization of α7 KD hippocampal neuronal cultures (**Bv**–**viii**), co-transduced with GFP-labelled *MEN1* encoding AAV (**Bv**) compared to scrambled controls, (**Bi**–**iv**) co-transduced with GFP only AAV, 4 weeks post-surgery (*n* = 35 images, six independent samples, representative image). tdTomato-positive neurons (**Bii**,**vi**) and GFP-positive neurons (**Bi**,**v**) labelled with α-C-terminal menin (**Biii**,**vii**) and merged (**Biii**,**vi**), Bvii exhibits significantly increased expression of C-menin in tdTomato GFP-positive hippocampal-pyramidal neurons compared to α7 KD+GFP only control (**Biii**). (**C**) Summary data, normalized menin expression in neurons from α7 nAChR scrambled AAV, α7 nAChR shRNA AAV, α7 nAChR scrambled AAV+*MEN1*, α7 nAChR AAV+*MEN1*AAV and α7 nAChR AAV+GFP AAV neuronal cultures compared to relevant controls (*n* ≥ 25 images, ≥7 independent samples) DIV 3, 7, 10, 14 and 20. (**D**) Summary data, normalized menin expression in neurons from α7 nAChR scrambled AAV, α7 nAChR shRNA AAV, α7 nAChR scrambled AAV+*MEN1*, α7 nAChR AAV+*MEN1*AAV and α7 nAChR AAV+GFP AAV compared to relative controls (*n* ≥ 30 images, ≥6 independent samples, 8 weeks old). Scale bars, (**Ai**–**iii**) 10, (**Aiv**–**vi**) 10, (**Bi**–**iv**) 5 and (**Bv**–**viii**) 8 μm. White dotted circles indicate GFP-positive pyramidal neurons. White asterisks on slices indicate colocalization of NFl and GFP. White asterisks on neurites indicate menin expression in tdTomato+ GFP-positive neurites. Statistical significance (one-way ANOVA followed by Tukey’s multiple comparison test) (one sample *t*-test) **** *p* < 0.0001, *** *p* < 0.001, ** *p* < 0.01, ns *p* > 0.9999. See Appendix A.

**Figure 9 cells-10-03286-f009:**
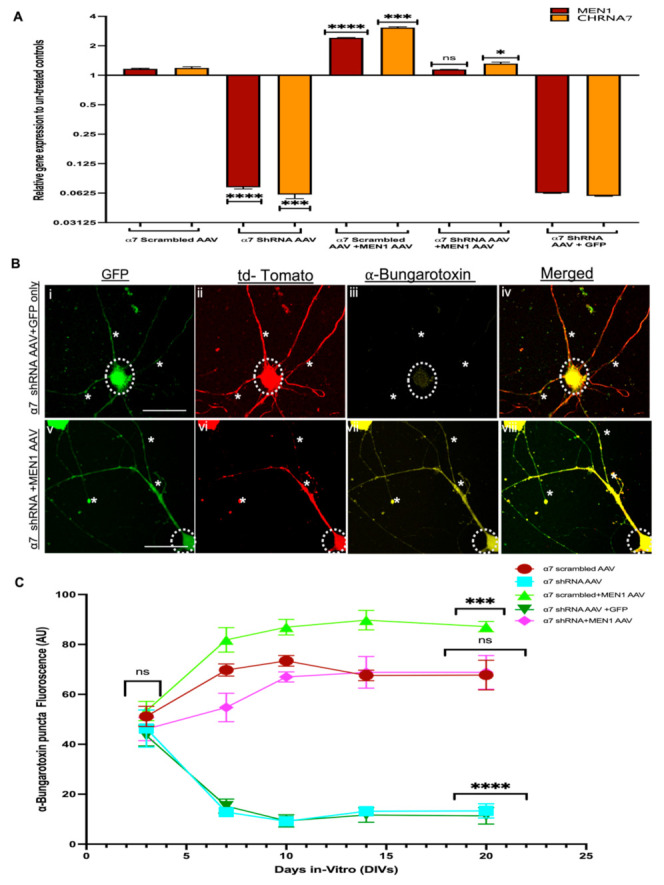
Overexpression of neuron-specific menin restores α7 nAChRs puncta expression in α7 KD hippocampal neurons (in vitro). (**A**) Summary data, fold change gene expression in from α7 nAChR scrambled AAV, α7 nAChR shRNA AAV, α7 nAChR scrambled AAV+*MEN1*, α7 nAChR AAV+*MEN1*AAV and α7 nAChR AAV+GFP AAV hippocampal neuronal cultures relative to untreated controls, determined by qPCR (*n* = 6, three independent experiments each, DIV 20). Upregulation of *MEN1* and CHRNA7 genes was observed in α7 nAChR scrambled AAV+*MEN1*, whereas the restoration of *MEN1* and CHRNA7 was observed in α7 nAChR AAV+*MEN1*AAV neuronal samples. (**B**) ICC characterization of α7 KD+*MEN1* AAV hippocampal neuronal cultures (**Bv**–**viii**), co-transduced with GFP-labelled *MEN1* encoding AAV (**Bv**) compared to scrambled controls (**Bi**-**iv**) co-transduced with GFP only AAV (*n* = 25 images, six independent samples, representative image). tdTomato-positive neurons (**Bii**,**vi**) and GFP-positive neurons (**Bi**,**v**) labelled with α-bungarotoxin (**Biii**,**vii**) and merged (**Biii**,**vi**), (**Bvii**) exhibits significantly increased expression of α7 nAChRs in tdTomato+GFP-positive hippocampal pyramidal neurons compared to α7 KD+GFP only control (**Biii**). (**C**) Summary data, normalized bungarotoxin expression in neurons from α7 nAChR scrambled AAV, α7 nAChR shRNA AAV, α7 nAChR scrambled AAV+*MEN1,* α7 nAChR AAV+*MEN1*AAV and α7 nAChR AAV+GFP AAV neuronal cultures compared to relevant controls (*n* ≥ 35 images, ≥7 independent samples) DIV 3, 7, 10, 14 and 20.Scale bars, (**Bi**–**iv**) 15 and (**Biv**–**vi**) 12 μm. White dotted circles indicate GFP-positive+tdTomato-positive pyramidal neurons. White asterisks on neurites indicate α7 nAChRs expression in tdTomato+ GFP-positive neurites. Statistical significance (one-way ANOVA followed by Tukey’s multiple comparison test) **** *p* < 0.0001, *** *p* < 0.001, * *p* < 0.1, ns *p* > 0.9999. See Appendix A.

**Table 1 cells-10-03286-t001:** DNA sequence corresponding to the shRNA sequences used for RNAi KD experiments.

Gene	Genbank Accession Number	97 Mer shRNA Sequence (siRNA Underlined)	Splash Score	Antisense. Guide. Sequence
*CHRNA7*	NM_007390	TGCTGTTGACAGTGAGCGAGACCAGGATCATTCTTCTGAATAGTGAAGCCACAGATGTATTCAGAAGAATGATCCTGGTCCTGCCTACTGCCTCGGA	1.49	TTCAGAAGAATGATCCTGGTCC
α7 scrambled	NM_007390	TGCTGTTGACAGTGAGCGATCAACTATATGCGAGGTGACTTAGTGAAGCCACAGATGTAAGTCACCTCGCATATAGTTGAGTGCCTACTGCCTCGGA	1.49	AGTCACCTCGCATATAGTTGAG

## Data Availability

The data presented in this study are available on request from the corresponding author.

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
