# Peer review of "Neuronal Menin Overexpression Rescues Learning and Memory Phenotype in CA1-Specific α7 nAChRs KD Mice"

_cells, 2021, doi:10.3390/cells10123286_

Round 1
Reviewer 1 Report
The manuscript by Batool et al. reports several correlations between menin expression and alpha7 nicotinic receptors. However, the numerous interesting observation are primarily correlational rather than clearly indicative of causal relationships. Many observations reflect transient changes, things go up and then down without any concept of why. It seems to me that some controls may be missing.
They do provide a large amount of data but, at least to me, it does not entirely come together as a compelling story. They remark that "menin, the protein product encoded by MEN1 gene has been extensively studied for its role as a tumor suppressor" but don't give the readers enough appreciation of this previous work to understand if the role of menin as a tumor suppressor might indeed underlie some of the observations reported.
Reading the manuscript is hampered by the presentation of the data. Specifically, the keys and axis labels of most of the figures are too small.
Other comments:
The abbreviation IHC is used on page 2 before it is explained on page 3.
PFA is introduced as an abbreviation for paraformaldehyde but never used again, and so it should be omitted.
OCT is apparently used as an abbreviation but never defined.
Many of the literature citations in the introduction are outdated review articles rather than current original research papers.
The reference format is unpleasant, with all but the first author indicated by just initials. It does not seem correct for the journal and is not even used consistently. References 27, 28, and 90 are different.
Author Response
We are grateful to this referee for his/her constructive comments and the provided feedback. Most issues raised were fair and this manuscript is now revised in light of all the referees’ suggestions. The concerns raised by the reviewer 1 are addressed bellow on a point-by-point basis.
Reviewer 1:
The manuscript by Batool et al. reports several correlations between menin expression and alpha7 nicotinic receptors. However, the numerous interesting observations are primarily correlational rather than clearly indicative of causal relationships. Many observations reflect transient changes, things go up and then down without any concept of why. It seems to me that some controls may be missing.
Response: Indeed, the referee is correct in stating that the genes studied here show a causal relationship, which is likely correlative. The issue raised by the referee is fair but the answer to the question raised, reside in previous studies which have demonstrated a clear relationship between MEN1 gene and the nAChRs. This study builds on that previous work[1-4]. It is not unusual for genes to exhibit such up and down regulation, as the networks of gene with cognate functions do indeed show a direct correlation and an interdependence. The evidence provided in the current study using both in-vitroand in-vivo approaches is another example of such interdependence. We have demonstrated that MEN 1 gene and its product which targets nAChRs to the synaptic sites exhibits a direct relationship with these cholinergic receptors. We have demonstrated that when nAChRs are knocked down, a lack of receptor function triggers an upregulation of menin to ensure that a deficit resulting from receptor absence is compensated via the targeting of the existing receptors. This is what has been demonstrated in other systems as well and that literature is cited in this paper. We next demonstrated that in the absence of a persistent and ensuing receptor activity, menin expression is downregulated, which then culminates into behavioural deficit that we have reported here. In this context, we have included all possible controls which are now specifically highlighted in the paper for this referee’s referral.
They do provide a large amount of data but, at least to me, it does not entirely come together as a compelling story. They remark that "menin, the protein product encoded by MEN1 gene has been extensively studied for its role as a tumor suppressor" but don't give the readers enough appreciation of this previous work to understand if the role of menin as a tumor suppressor might indeed underlie some of the observations reported.
The referee has raised an interesting point; however, the goal of this study was to highlight and explore the role of menin as a synaptogenic molecule, which we were the first group to demonstrate, rather than as a tumour suppressor. We have however, highlighted a section in the discussion where menin’s role as a signalling molecule in the light of our findings has been discussed. It is important to note that even though MEN1 gene was identified as a tumor suppressor, since its original role it has been discovered to serve myriad functions – synapse formation and synaptic plasticity being now most extensively studied.
Reading the manuscript is hampered by the presentation of the data. Specifically, the keys and axis labels of most of the figures are too small.
We acknowledge the reviewer’s concerns, all figures have now been changed and the axis are adjusted accordingly.
Other comments:
The abbreviation IHC is used on page 2 before it is explained on page 3.
This has now been corrected (page 2) and have highlighted in yellow.
PFA is introduced as an abbreviation for paraformaldehyde but never used again, and so it should be omitted.
Changes made as per the suggestion.
OCT is apparently used as an abbreviation but never defined.
The abbreviation has been defined.
Many of the literature citations in the introduction are outdated review articles rather than current original research papers.
Updated research articles references have been added in the introduction section as per the suggestion.
The reference format is unpleasant, with all but the first author indicated by just initials. It does not seem correct for the journal and is not even used consistently. References 27, 28, and 90 are different.
MDPI style of references has been used to edit the references as per the suggestion.
- Batool, S.a.Z.J.a.A.B.a.U.A.K.a.V.F.a.S.N.I. Spatiotemporal Patterns of Menin Localization in Developing Murine Brain: Co-Expression with the Elements of Cholinergic Synaptic Machinery. Cells (Basel, Switzerland) 2021, 10, 1215.
- Flynn, N.a.G.A.a.V.F.a.J.T.A.a.S.N.I. Menin: A Tumor Suppressor That Mediates Postsynaptic Receptor Expression and Synaptogenesis between Central Neurons of Lymnaea stagnalis.(Research Article). PLoS ONE 2014, 9.
- Getz, A.M.a.V.F.a.B.E.M.a.X.F.a.F.N.M.a.Z.W.a.S.N.I. Two proteolytic fragments of menin coordinate the nuclear transcription and postsynaptic clustering of neurotransmitter receptors during synaptogenesis between Lymnaea neurons. Scientific reports 2016, 6, 31779--31779.
- van Kesteren, R.E.a.S.N.I.a.M.D.W.a.B.J.a.F.Z.-P.a.G.W.P.M.a.S.A.B. Synapse Formation between Central Neurons Requires Postsynaptic Expression of the MEN1 Tumor Suppressor Gene. The Journal of neuroscience 2001, 21, 161--RC161.
Reviewer 2 Report
The manuscript by Batool and colleagues investigates the potential role of menin overexpression in rescuing cognitive capabilities in CA1-specific α7 nAChRs KD mice. They present proof that overexpression of menin rescues α7 nAChRs KD functional and behavior deficits, specifically in hippocampal (CA1) neurons. The study documented that knocking out of α7 nAChRs initially upregulates and then downregulates menin expressions in vitro and in vivo. Overall, the studies are well-conducted, and the data are novel and of interest. The authors did not overinterpret their results.
Comments: My overall suggestion to authors would be to, where that is possible, to combine figures. Some of the figures could be combined into one figure, and it would improve the manuscript. Some of the axis legends are hard to read (fig 1c, d).
- In Fig 10 C, D, the authors left out α7 scrambled AAV MEN1 group on both graphs. If they did not want to incorporate this group into the main figure, they must at least compare group α7 scrambled AAV to α7 scrambled AAV MEN1 group so that readers can see those results. Positions of the statistical significance bars in Fig 10 D are confusing, and maybe they should be moved on a left x-axis
- This comment goes along with the previous one, there is no supplementary table 19 (s19). Maybe this also could be a place to compare two previously mentioned groups.
Specific comments:
P5, line 217. – Please put in the volume of AAV virus that was injected bilaterally
P28 – Figure 10 is also labeled as Fig 9
Author Response
We are grateful to the reviewers for their constructive comments and the feedback that they have provided. Most issues raised were fair and this manuscript is now revised in light of all the referee’s suggestions. The concerns raised by this referee are addressed bellow on a point-by-point basis.
My overall suggestion to authors would be to, where that is possible, to combine figures. Some of the figures could be combined into one figure, and it would improve the manuscript. Some of the axis legends are hard to read (fig 1c, d).
Response: As per the referee’s suggestions, the figures have been merged where appropriate, and the axis legends have been tailored accordingly.
In Fig 10 C, D, the authors left out α7 scrambled AAV MEN1 group on both graphs. If they did not want to incorporate this group into the main figure, they must at least compare group α7 scrambled AAV to α7 scrambled AAV MEN1 group so that readers can see those results. Positions of the statistical significance bars in Fig 10 D are confusing, and maybe they should be moved on a left x-axis. This comment goes along with the previous one, there is no supplementary table 19 (s19). Maybe this also could be a place to compare two previously mentioned groups.
Response: The missing data is now added to the freezing percentage graph which is now moved to the supplementary Data section. The table S18 has been updated to address the missing data as suggested by the reviewer. The typo error for S19 is now corrected in the manuscript.
- P5, line 217. – Please put in the volume of AAV virus that was injected bilaterally
Response: The volume of the AAV injected is highlighted in yellow.
- P28 – Figure 10 is also labeled as Fig 9
Response: The error has been corrected
Reviewer 3 Report
This is an interesting and well-done study showing a reciprocal relationship between a7 nAChRs and menin in the hippocampus. The general idea is clear; some details need to be explained, as follows.
- It is known that neuronal a7 nAChRs are located mainly extrasynaptically. How this fits with the authors findings?
- It is not clear how menin overexpression can rescue a7 expression in a7 KD cells.
- Paragraph 734-755 is unclear; lines 734-738 should be edited.
Author Response
We are grateful to the reviewers for their constructive comments and the feedback that they have provided. Most issues raise were fair and this manuscript is now revised in light of the referee’s suggestions. The concerns raised by this referee are addressed bellow on a point-by-point basis.
- It is known that neuronal a7 nAChRs are located mainly extrasynaptically. How this fits with the authors findings?
Response: The a 7 nAChRs are located at the pre- and postsynaptic sites - both at the neurons and glial cells in the CNS (Tribollet et al., 2004). In this current study, we specifically knocked down a7 nAChRs in neurons (which includes both synaptic and non-synaptic a 7 nAChRs). We have shown previously both in vivo and vitro that the a 7nAChRs are located at specific synaptic sites along with other elements of the synaptic machinery at the glutamatergic hippocampal neurons. Using an shRNA knockdown approach, we have shown that when menin expression is altered that the nAChRs receptor expression levels are also perturbed. These previous studies have thus provided unequivocal evidence showing a direct relationship between menin and the nAChRs at the synaptic sites. This study was however, specifically designed to look at the interdependence of a 7 nAChRs with menin with respect to the synaptic assembly and the function. We have added a section in the discussion which considers the other cells where a 7 nAChRs could also be manipulated in future to study the interdependence of the two proteins.
- It is not clear how menin overexpression can rescue a7 expression in a7 KD cells.
Response: The reviewer has raised an interesting point regarding the mechanism underlying menin induced a7 nAChRs rescue in neurons. There are a number of possibilities that can be speculated to deduce the mechanism underlying our findings, which are discussed and highlighted in discussion (line 755-776). However, these are only speculations which warrant further study to dissect the precise mechanisms underlying the rescue but are beyond the scope of the present study.
- Paragraph 734-755 is unclear; lines 734-738 should be edited.
Response: The sentences have been restructured as per the referee’s suggestion.